# Learning from Few Samples: Transformation-Invariant SVMs with Composition and Locality at Multiple Scales

**Tao Liu[1], P. R. Kumar[1], Ruida Zhou[1], Xi Liu[2]**
[1]Texas A&M University, College Station, TX, USA
[2]Applied Machine Learning, Facebook AI, Menlo Park, CA, USA
{tliu, prk, ruida}@tamu.edu, xliu1@meta.com

## Abstract

Motivated by the problem of learning with small sample sizes, this paper shows how to incorporate into support-vector machines (SVMs) those properties that have made convolutional neural networks (CNNs) successful. Particularly important is the ability to incorporate domain knowledge of invariances, e.g., translational invariance of images. Kernels based on the *maximum* similarity over a group of transformations are not generally positive definite. Perhaps it is for this reason that they have not been studied theoretically. We address this lacuna and show that positive definiteness indeed holds *with high probability* for kernels based on the maximum similarity in the small training sample set regime of interest, and that they do yield the best results in that regime. We also show how additional properties such as their ability to incorporate local features at multiple spatial scales, e.g., as done in CNNs through max pooling, and to provide the benefits of composition through the architecture of multiple layers, can also be embedded into SVMs. We verify through experiments on widely available image sets that the resulting SVMs do provide superior accuracy in comparison to well-established deep neural network benchmarks for small sample sizes.

## 1 Introduction

With the goal of learning when the number of training samples is small, and motivated by the success of CNNs [35, 31], we wish to endow SVMs with as much a priori domain knowledge as possible.

One such important domain property for image recognition is translational invariance. An image of a dog remains an image of the same dog if the image is shifted to the left. Similarly, it is also rotation-invariant. More generally, given a group of transformations under which the classification of images is invariant, we show how to endow SVMs with the knowledge of such invariance.

One common approach is data augmentation [29, 6], where several transformations of each training sample are added to the training set. When applying SVMs on this augmented training set, it corresponds to a kernel that defines the similarity between two vectors $X_1$ and $X_2$ as the *average similarity* between $X_1$ and all transformations of $X_2$. However the average also includes transformations that are maximally dissimilar, and we show that it leads to poor margins and poor classification results. More appealing is to construct a kernel that defines similarity as the *maximum similarity* between $X_1$ and all transformations of $X_2$. We show that this kernel is positive definite with high probability in the small sample size regime of interest to us, under a probabilistic model for features. We verify this property on widely available datasets and show that the improvement obtained by endowing SVMs with this transformation invariance yields considerably better test accuracy.

36th Conference on Neural Information Processing Systems (NeurIPS 2022).

Another important domain property for image recognition is the "locality" of features, e.g., an edge depends only on a sharp gradient between neighboring pixels. Through operations such as max-pooling, CNNs exploit locality at multiple spatial scales. We show how one may incorporate such locality into polynomial SVMs.

Finally, their multi-layer architecture provides CNNs the benefits of composition [7]. We show how one can iteratively introduce multiple layers into SVMs to facilitate composition. The introduction of multiple layers increases computational complexity, and we show how this can be alleviated by parallel computation so as to achieve a reduction of computation time by increasing memory.

We show experimentally that the resulting SVMs provide significantly improved performance for small datasets. Translational and rotational invariance embedded into SVMs allows them to recognize objects that have not already been centered in images or oriented in upright positions; referred to as transformed datasets in the sequel. The transformation-invariant SVMs provide significant improvements over SVMs as well as CNN benchmarks without data augmentation when the training set is small. For 100/200/500 training samples, the recognition accuracy of the MNIST dataset [17] is increased, respectively, from the figures of 68.33%/83.20%/91.33% reported by the CNNs optimized over architectures and dimensions [10] to, respectively, 81.55%/89.23%/93.11%. Similar improvements are also obtained in the EMNIST Letters [4] and Transformed MNIST datasets. Computational results reported here are for small datasets, handled efficiently by LIBSVM [2].

## 1.1   Background

In the early 2000s, SVMs [5] were one of the most effective methods for image classification [19]. They had a firm theoretical foundation of margin maximization [32, 1] that is especially important in high dimensions, and a kernel method to make high-dimensional computation tractable. However, with the advent of CNNs and the enhanced computing power of GPUs, SVMs were replaced in image classification. One reason was that CNNs were able to incorporate prior knowledge. The pioneering paper [35] emphasizes that "It is usually accepted that good generalization performance on real-world problems cannot be achieved unless some a priori knowledge about the task is built into the system. Back-propagation networks provide a way of specifying such knowledge by imposing constraints both on the architecture of the network and on its weights. In general, such constraints can be considered as particular transformations of the parameter space." It further mentions specifically that, "Multilayer constrained networks perform very well on this task when organized in a hierarchical structure with shift-invariant feature detectors." Indeed, CNNs have successfully incorporated several important characteristics of images. One, mentioned above (called shift-invariance), is translational invariance, which is exploited by the constancy, i.e., location independence, of the convolution matrices. A second is locality. For example, an "edge" in an image can be recognized from just the neighboring pixels. This is exploited by the low dimensionality of the kernel matrix. A third characteristic is the multiplicity of spatial scales, i.e., a hierarchy of spatial "features" of multiple sizes in images. These three properties are captured in modern CNNs through the "pooling" operation at the $(\ell + 1)$-th layer, where the features of the $\ell$-th layer are effectively low-pass filtered through operations such as max-pooling. More recently, it has been shown that depth in neural networks (NNs) of rectified linear units (ReLUs) permits composition, enhances expressivity for a given number of parameters, and reduces the number needed for accurate approximations [7].

Generally, neural networks have become larger over time with the number of parameters ranging into hundreds of millions, concomitantly also data-hungry, and therefore inapplicable in applications where data is expensive or scarce, e.g., medical data [20]. For these reasons, there is an interest in methodologies for learning efficiently from very few samples, which is the focus of this paper.

## 1.2   Related Work

There are mainly two sets of studies: on transformation-invariant kernels and on local correlations.

**Transformation-Invariant Kernels.** There are two major routes to explore transformation invariant kernels [26, 16]. The most widely used is based on Haar-integration kernels, called "average-fit" kernels in this paper, which average over a transformation group [27, 12, 11, 23, 21, 22]. Although trivially positive definite, they appear to produce poor results in our testing (Section 4). Mairal et al. [21] reported some good results when a large dataset is available for pre-training bottom network layers unsupervised [24], but our focus is on data-scarce situations where no such large dataset is

available. This paper concentrates on "best-fit" kernels, which are based on the maximum similarity over transformations. "Jittering kernels" [8, 9] calculate the minimum distance between one sample and all jittered forms of another sample, analogous to this paper. Instead of measuring the minimum distance between samples, tangent distance kernels [30, 13] use a linear approximation to measure the minimum distance between sets generated by the transformation group, which can only guarantee local invariance. In comparison, our "best-fit" kernel is defined as the maximum inner product between one sample and all transformed forms of another sample, which enjoys global invariance and differs from jittering kernels for non-norm-preserving transformations (e.g., scaling transformations). Although "best-fit" kernels enjoy a good performance, they are not guaranteed to be positive definite, thus the global optimality of the SVM solution cannot be guaranteed theoretically. We address this lacuna and show that positive definiteness indeed holds *with high probability* in the small training sample set regime for the "best-fit" kernel. Additionally, we show that locality at multiple scales can further improve the performance of "best-fit" kernels. We note that there were also some works that treat an indefinite kernel matrix as a noisy observation of some unknown positive semidefinite matrix [3, 36], but our goal is to analyze conditions that make the "best-fit" kernel positive definite.

**Local Correlations.** Since "local correlations," i.e., dependencies between nearby pixels, are more pronounced than long-range correlations in natural images, Scholkopf et al. [25] defined a two-layer kernel utilizing dot product in a polynomial space which is mainly spanned by local correlations between pixels. We extend the structure of two-layer local correlation to multilayer architectures by introducing further compositions, which gives the flexibility to consider the locality at multiple spatial scales. We also analyze the corresponding time and space complexity of multilayer architectures.

## 2 Kernels with Transformational Invariance

To fix notation, let $\mathcal{S} = \{(X_1, y_1), \ldots, (X_n, y_n)\}$ be a set of $n$ labeled samples with $m$-dimensional feature vectors $X_i = (X_i^{(1)}, \ldots, X_i^{(m)})^T \in \mathcal{X} \subset \mathbb{R}^m$ and labels $y_i \in \mathcal{Y}$.

### 2.1 Transformation Groups and Transformation-Invariant Best-Fit Kernels

We wish to endow the kernel of the SVM with the domain knowledge that the sample classification is invariant under certain transformations of the sample vectors. Let $\mathcal{G}$ be a transformation group that acts on $\mathcal{X}$, i.e., for all $S, T, U \in \mathcal{G}$: (i) $T$ maps $\mathcal{X}$ into $\mathcal{X}$; (ii) the identity map $I \in \mathcal{G}$; (iii) $ST \in \mathcal{G}$; (iv) $(ST)U = S(TU)$; (v) there is an inverse $T^{-1} \in \mathcal{G}$ with $TT^{-1} = T^{-1}T = I$.

We start with a base kernel $K_{base} : \mathcal{X} \times \mathcal{X} \to \mathcal{R}$ that satisfies the following properties:

1. Symmetry, i.e., $K_{base}(X_i, X_j) = K_{base}(X_j, X_i)$;
2. Positive definiteness, i.e., positive semi-definiteness of its Gram matrix: $\sum_{i=1}^{n} \sum_{j=1}^{n} \alpha_i \alpha_j K_{base}(X_i, X_j) \geq 0$ for all $\alpha_1, \ldots, \alpha_n \in \mathbb{R}$;
3. $K_{base}(TX_i, X_j) = K_{base}(X_i, T^{-1}X_j), \forall T \in \mathcal{G}$.

Define the kernel with the "best-fit" transformation over $\mathcal{G}$ by

$$K_{\mathcal{G}, best, base}(X_i, X_j) := \sup_{T \in \mathcal{G}} K_{base}(TX_i, X_j). \tag{1}$$

**Lemma 1.** $K_{\mathcal{G}, best, base}$ *is a symmetric kernel that is also transformation-invariant over the group* $\mathcal{G}$, *i.e.,* $K_{\mathcal{G}, best, base}(TX_i, X_j) = K_{\mathcal{G}, best, base}(X_i, X_j)$.

*Proof.* The symmetry follows since

$$K_{\mathcal{G}, best, base}(X_i, X_j) = \sup_{T \in \mathcal{G}} K_{base}(TX_i, X_j) = \sup_{T \in \mathcal{G}} K_{base}(X_i, T^{-1}X_j)$$

$$= \sup_{T \in \mathcal{G}} K_{base}(T^{-1}X_j, X_i) = \sup_{T^{-1} \in \mathcal{G}} K_{base}(T^{-1}X_j, X_i) = K_{\mathcal{G}, best, base}(X_j, X_i).$$

The transformational invariance follows since

$$K_{\mathcal{G}, best, base}(TX_i, X_j) = \sup_{S \in \mathcal{G}} K_{base}(STX_i, X_j) = \sup_{UT^{-1} \in \mathcal{G}} K_{base}(UX_i, X_j)$$

$$= \sup_{U \in \mathcal{G}} K_{base}(UX_i, X_j) = K_{\mathcal{G}, best, base}(X_i, X_j).$$

*The Translation Group:* Of particular interest in image classification is the group of translations, a subgroup of the group of transformations. Let $X_i = \{X_i^{(p,q)} : p \in [m_1], q \in [m_2]\}$ denote a two-dimensional $m_1 \times m_2$ array of pixels, with $m = m_1 m_2$. Let $T_{rs} X_i := \{X_i^{((p-r) \bmod m_1, (q-s) \bmod m_2)} : p \in [m_1], q \in [m_2]\}$ denote the transformation that translates the array by $r$ pixels in the $x$-direction, and by $s$ pixels in the $y$-direction. The translation group is $\mathcal{G}_{trans} := \{T_{rs} : r \in [m_1], s \in [m_2]\}$. For notational simplicity, we will denote the resulting kernel $K_{\mathcal{G}_{trans},best,base}$ by $K_{TI,best,base}$.

## 2.2 Positive Definiteness of Translation-Invariant Best-Fit Kernels

There are two criteria that need to be met when trying to embed transformational invariance into SVM kernels. (i) The kernel will need to be invariant with respect to the particular transformations of interest in the application domain. (ii) The kernel will need to be positive definite to have provable guarantees of performance.

$K_{TI,best,base}$ satisfies property (i) as established in Lemma 1. Concerning property (ii) though, in general, $K_{TI,best,base}$ is an indefinite kernel. We now show that when the base kernel is a normalized linear kernel, $K_{linear}(X_i, X_j) := \frac{1}{m} X_i^T X_j$, then it is indeed positive definite in the small sample regime of interest under a probabilistic model for dependent features.

**Assumption 1.** Suppose that $n$ samples $\{X_1, X_2, \ldots, X_n\}$ are i.i.d., with $X_i = \{X_i^{(1)}, X_i^{(2)}, \ldots, X_i^{(m)}\}$ being a normal random vector with $X_i \sim \mathcal{N}(0, \Sigma), \forall i \in [n]$. Suppose also that $\|\lambda\|_2 / \|\lambda\|_\infty \geq (\ln m)^{\frac{1+\epsilon}{2}} / 2$ for some $\epsilon \in (0, 1]$, where $\lambda := (\lambda^{(1)}, \ldots, \lambda^{(m)})$ is comprised of the eigenvalues of $\Sigma$. Note that $X_i^{(p)}$ may be correlated with $X_i^{(q)}$, for $p \neq q$.

**Example 1.** *When $\Sigma = I_m$, i.e., $X_i^{(p)} \sim \mathcal{N}(0, 1), \forall p \in [m]$, the condition $\|\lambda\|_2 / \|\lambda\|_\infty \geq (\ln m)^{\frac{1+\epsilon}{2}} / 2$ holds trivially since $\|\lambda\|_2 = \sqrt{m}$ and $\|\lambda\|_\infty = 1$. We note that for independent features, we can relax the normality (Assumption 1) to sub-Gaussianity.*

**Theorem 1.** *Let*

$$K_{TI,best,linear}(X_i, X_j) := \sup_{T \in \mathcal{G}_{trans}} \frac{1}{m} (TX_i)^T X_j \tag{2}$$

*be the best-fit translation invariant kernel with the base kernel chosen as the normalized linear kernel. Under Assumption 1, if $n \leq \frac{\|\lambda\|_1}{2\|\lambda\|_2 (\ln m)^{\frac{1+\epsilon}{2}}}$, then $K_{TI,best,linear}$ is a positive definite kernel with probability approaching one, as $m \to \infty$.*

*Outline of proof.* We briefly explain ideas behind the proof, with details presented in Appendix B. For brevity, we denote $K_{TI,best,linear}(X_i, X_j)$ and $K_{linear}(X_i, X_j)$ by $K_{TI,ij}$ and $K_{ij}$, respectively. From Gershgorin's circle theorem [33] every eigenvalue of $K_{TI,best,linear}$ lies within at least one of the Gershgorin discs $\mathcal{D}(K_{TI,ii}, r_i) := \{\lambda \in \mathbb{R} \mid |\lambda - K_{TI,ii}| \leq r_i\}$, where $r_i := \sum_{j \neq i} |K_{TI,ij}|$. Hence if $K_{TI,ii} > \sum_{j \neq i} |K_{TI,ij}|, \forall i$, then $K_{TI}$ is a positive definite kernel. Accordingly, we study a tail bound on $\sum_{j \neq i} |K_{Ti,ij}|$ and an upper bound on $K_{TI,ii}$, respectively, to complete the proof. □

Note that $\|\lambda\|_1 \leq \sqrt{m} \|\lambda\|_2$ for an $m$-length vector $\lambda$, which implies that $n = \tilde{O}(\sqrt{m})$.

We now show that positive definiteness in the small sample regime also holds for the polynomial kernels which are of importance in practice:

$$K_{poly}(X_i, X_j) := (1 + \frac{\gamma}{m} X_i^T X_j)^d \text{ for } \gamma \geq 0 \text{ and } d \in \mathbb{N}.$$

**Theorem 2.** *For any $\gamma \in \mathbb{R}_+$ and $d \in \mathbb{N}$, the translation-invariant kernels,*

$$K_{TI,best,poly}(X_i, X_j) := \sup_{T \in \mathcal{G}_{trans}} (1 + \frac{\gamma}{m} (TX_i)^T X_j)^d \tag{3}$$

*are positive definite with probability approaching one as $m \to +\infty$, under Assumption 1, when $\|\lambda\|_\infty \leq 1$, and $n \leq \frac{\|\lambda\|_1}{2\|\lambda\|_2 (\ln m)^{\frac{1+\epsilon}{2}}}$.*

Table 1: The value of $n$ up to which the kernel is positive definite. Positive definiteness continues to hold for moderate sample sizes, indicating that the theorem is conservative.

| Datasets | $K_{TI,best,linear}$ | $K_{TI,best,poly}$ |
|---|---|---|
| Original MNIST | $\approx 45$ | $\approx 375$ |
| EMNIST | $\approx 35$ | $\approx 395$ |
| Translated MNIST | $\approx 455$ | $\approx 15000$ |

*Proof.* Note that $\|\lambda\|_\infty \leq 1$ can be satisfied simply by dividing all entries of the $X_i$s by $\sqrt{\|\lambda\|_\infty}$. The detailed proof is presented in Appendix C. $\square$

**Remark.** Since Gershgorin's circle theorem is a conservative bound on the eigenvalues of a matrix, the bound $n = \tilde{O}(\sqrt{m})$ on the number of samples for positive definiteness is also conservative. In practice, positive definiteness of $K_{TI,best,linear}$ holds for larger $n$. Even more usefully, $K_{TI,best,poly}$ is positive definite for a much larger range of $n$ than $K_{TI,best,linear}$, as reported in Table 1.

## 2.3 Comparison with the Average-Fit Kernel and Data Augmentation

### 2.3.1 Average-Fit Kernel

In [12], the following "average-fit kernel" kernel is considered

$$K_{\mathcal{G}_{trans},avg,linear}(X_i, X_j) := \frac{1}{|\mathcal{G}|} \sum_{T \in \mathcal{G}} K_{linear}(TX_i, X_j), \tag{4}$$

which seeks the "average" fit over all transformations. We denote it by $K_{TI,avg,linear}(X_i, X_j)$ for short. It is trivially invariant with respect to the transformations in $\mathcal{G}$ and positive definite. However, it is not really a desirable choice for translations when the base kernel is the linear kernel. Note that $\frac{1}{|\mathcal{G}_{trans}|} \sum_{T \in \mathcal{G}_{trans}} TX_i = \alpha(1, 1, \ldots, 1)^T$, where $\alpha = \frac{1}{m} \sum_{p \in [m_1], q \in [m_2]} X_i^{(p,q)}$ is the average brightness level of $X_i$. Therefore $K_{TI,avg,linear}(X_i, X_j) = m \times$ (Avg brightness level of $X_i$) $\times$ (Avg brightness level of $X_j$). The kernel solely depends on the average brightness levels of the samples, basically blurring out all details in the samples. In the case of rotational invariance, it depends only on the average brightness along each concentric circumference. As expected, it produces very poor results, as seen in the experimental results in Section 4.

### 2.3.2 Data Augmentation

Data augmentation is a popular approach to learn how to recognize translated images. It augments the dataset by creating several translated copies of the existing samples. (A special case is the virtual support vectors method which augments the data with transformations of the support vectors and retrains [26].) SVM with kernel $K_{base}$ applied to the augmented data is equivalent to employing the average-fit kernel as (4). Consider the case where the augmented data consists of all translates of each image. The resulting dual problem for SVM margin maximization [5] is:

$$\max_\lambda \ -\frac{1}{2} \sum_{i,j,T_1,T_2} \lambda_{i,T_1} \lambda_{j,T_2} y_i y_j K_{base}(T_1 X_i, T_2 X_j) + \sum_{i,T} \lambda_{i,T}$$

$$\text{s.t. } \lambda_{i,T} \geq 0, \ \forall i \in [n], \forall T \in \mathcal{G}_{trans}; \ \sum_{i,T} \lambda_{i,T} y_i = 0.$$

The corresponding classifier is $\text{sign}(\sum_{i,T} \lambda_{i,T}^* y_i K_{base}(TX_i, X) + b^*)$, where $\lambda^*$ is the optimal dual variable and $b^* = y_j - \sum_{i,T} \lambda_{i,T}^* y_i K_{base}(TX_i, T'X_j)$, for any $j$ and $T'$ satisfying $\lambda_{j,T'}^* > 0$. When no data augmentation is implemented, i.e., $|\mathcal{G}_{trans}| = 1$, we use $\lambda_i$ as shorthand for $\lambda_{i,1}$. As shown in Theorem 4.1 [18], this is simply the dual problem for the SVM with $K_{TI,avg,base}$, and $\sum_i \lambda_i K_{TI,avg,base}(X_i, X_j) = \sum_{i,T \in \mathcal{G}} \lambda_{i,T} K_{base}(TX_i, X_j) \forall j$. Hence data augmentation is mathematically equivalent to a kernel with average similarity over all transformations. This yields a poor classifier since it only leverages the average brightness level of an image.

A simple example illustrates superiority of $K_{TI,best,linear}$ over $K_{TI,avg,linear}$ or data augmentation.

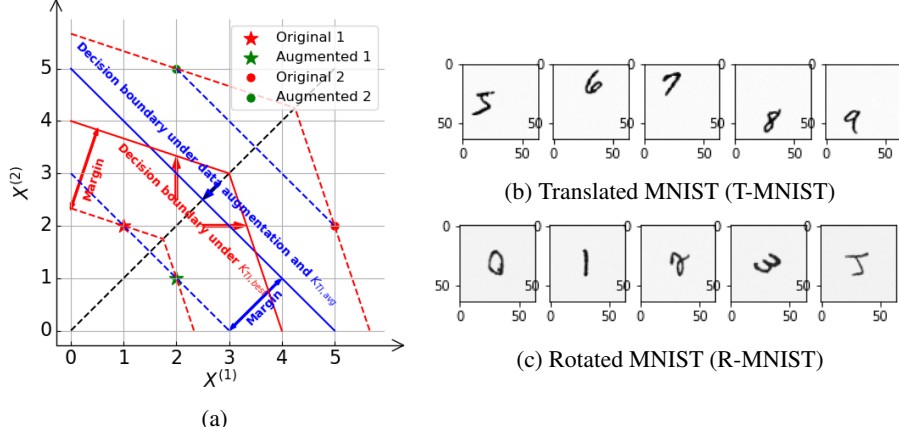

(a)

(b) Translated MNIST (T-MNIST)

(c) Rotated MNIST (R-MNIST)

Figure 1: (a) The kernel $K_{TI,best,linear}$ produces the piecewise linear separatrix shown in red. It yields a larger margin than the blue linear separatrix, i.e., decision boundary, that data augmentation and $K_{TI,avg}$ yield; (b) & (c) Transformed MNIST with random translation/rotation and Gaussian noise ($\mu = 0, \sigma = 0.1$).

**Example 2.** *Consider a special case of the translation group with $m_1 = 2$ and $m_2 = 1$. Consider a training set with just two samples $X_1 = (1, 2)$ and $X_2 = (5, 2)$, shown in red in Figure 1. Note that a translation in the image space is a right shift of vector elements. (So a translation of $X_1 = (1, 2)$ yields the augmented datum $X_3 = (2, 1)$ in this two-dimensional context because of wraparound.) Two new samples $X_3 = (2, 1)$ and $X_4 = (2, 5)$ are therefore generated through data augmentation, shown in green. The decision boundary of the base kernel with augmented data (equivalent to the average-fit kernel $K_{TI,avg}$) is shown by the blue solid line in Figure 1. Note that the linear decision boundary $X^{(1)} + X^{(2)} = 5$ depends solely on the brightness level $X^{(1)} + X^{(2)}$.*

*However, the decision boundary of the best-fit kernel $K_{TI,best,linear}(X_i, X_j) = \sup_{T \in \mathcal{G}_{trans}} \frac{1}{2}(TX_i)^T X_j$ is piecewise linear due to the "sup" operation. Each piece focuses only on the half of the samples that are on the same side of the symmetry axis (black dashed line), leading to the red piecewise linear separatrix with a larger margin. (The red margin is $\frac{\sqrt{10}}{2}$, which is larger than the blue margin $\sqrt{2}$.) The best-fit kernels thereby maximize the margin over the larger class of piecewise linear separatrices by exploiting the invariances. Even when data augmentation is used, it only produces linear separatrices and thus still has smaller margins. Benefits will be even more pronounced in higher dimensions. For other kernels (e.g., polynomial kernels), the shape of the decision boundary will be altered correspondingly (e.g., piecewise polynomial), but the TI best-fit kernel will still provide a larger margin.*

### 2.4 Rotation-Invariant Kernels

To mathematically treat rotation-invariant kernels, consider images that are circular, of radius $r$, with each concentric annulus from radius $\frac{(p-1)r}{m_1}$ to $\frac{pr}{m_1}$, for $p \in [m_1]$, comprised of $m_2$ pixels spanning the sectors $[2k\pi/m_2, 2(k + 1)\pi/m_2)$ for $k = 0, 1, \ldots, m_2 - 1$. Denote the element in the $p$-th annulus and $q$-th sector as "pixel" $X^{(p,q)}$, and define the rotation group $\mathcal{G}_{rotate} := \{T_1, T_2, ..., T_{m_2}\}$, where $T_{q'}X^{(p,q)} = X^{(p,q+q')}$. The rotation-invariant (RI) best-fit kernel is

$$K_{RI,best,base}(X_i, X_j) := \sup_{T \in \mathcal{G}_{rotate}} K_{base}(TX_i, X_j).$$

**Lemma 2.** *Under Assumption 1, the rotational invariant kernel $K_{RI,best,poly}$ is positive definite with probability approaching one as $m \to +\infty$, under the same conditions as in Theorem 1.*

In Section 4, we report on the large performance gain of $K_{RI,best,poly}$.

# 3 Incorporating Locality at Multiple Spatial Scales

To better illustrate the property of "locality" and its incorporation into SVMs, consider the simple context of a polynomial kernel and a one-dimensional real-valued pixel sequence.

Let us regard the individual pixel values in the sequence $\{X_i^{(1)}, X_i^{(2)}, \ldots, X_i^{(m)}\}$ as the primitive features at "Layer 0". Consider now a "local" feature depending only on the nearby pixels $\{X_i^{(\ell)}, X_i^{(\ell+1)}, \ldots, X_i^{(\ell+k_1)}\}$ that can be modeled by a polynomial of degree $d_1$. We refer to $k_1$ as the locality parameter.

Such a local feature is a linear combination of monomials of the form $\prod_{j=\ell}^{\min(\ell+k_1,m)} (X_i^{(j)})^{c_j}$ with $\sum_{j=\ell}^{\min(\ell+k_1,m)} c_j \leq d_1$ where each integer $c_j \geq 0$. This gives rise to a kernel

$$K_{L,ij}^{(1)} := \Big[ \sum_{\ell=1}^{m-k_1} \Big( \sum_{p=\ell}^{\ell+k_1} X_i^{(p)} X_j^{(p)} + 1 \Big)^{d_1} + 1 \Big]^{d_2}. \tag{5}$$

We regard "Layer 1" as comprised of such local features of locality parameter $k_1$, at most $k_1$ apart.

"Layer 2" allows the composition of local features at Layer 1 that are at most $k_2$ apart:

$$K_{L,ij}^{(2)} := \Big\{ \sum_{g=1}^{m-k_1-k_2} \Big[ \sum_{\ell=g}^{g+k_2} \Big( \sum_{p=\ell}^{\ell+k_1} X_i^{(p)} X_j^{(p)} + 1 \Big)^{d_1} 1 \Big]^{d_2} + 1 \Big\}^{d_3}. \tag{6}$$

This can be recursively applied to define deeper kernels with locality at several coarser spatial scales.

The above procedure extends naturally to two-dimensional images $\{X_i^{(p,q)} : p \in [m_1], q \in [m_2]\}$. Then the kernel at Layer 1 is simply $(\sum_{s=1}^{m_2-k_1} \sum_{\ell=1}^{m_1-k_1} (\sum_{q=s}^{s+k_1} \sum_{p=\ell}^{\ell+k_1} X_i^{(p,q)} X_j^{(p,q)} + 1)^{d_1} + 1)^{d_2}$. The resulting kernels are always positive definite:

**Lemma 3.** $K_L$ *is a positive definite kernel.*

*Proof.* Note that if $K_1$ and $K_2$ are positive definite kernels, then the following kernels $K$ obtained by Schur products [28], addition, or adding a positive constant elementwise, are still positive definite kernels: (i) $K_{ij} = \alpha K_{1,ij} + \beta K_{2,ij}$, (ii) $K_{ij} = (K_{1,ij})^{\ell_1} (K_{2,ij})^{\ell_2}$, (iii) $K_{ij} = K_{1,ij} + \gamma$, $\forall \alpha, \beta \geq 0, \ell_1, \ell_2 \in \mathbb{N}, \gamma \geq 0$. The kernel $K_L$ can be obtained by repeatedly employing the above operations with $\alpha = \beta = \gamma = 1$, starting with a base linear kernel which is positive definite with high probability under the conditions of Theorem 2. $\square$

One difference from CNNs is that, for the same input layer, one cannot have multiple output channels. If we design multiple channels with different degrees, then the channel with a larger degree will automatically subsume all terms generated by the channel with a smaller degree. Therefore, it is equivalent to having only one output channel with the largest degree. However, if the images have multiple channels to start with (as in R, G, and B, for example), then they can be handled separately. But after they are combined at a layer, there can only be one channel at subsequent higher layers.

**Combining Locality at Multiple Spatial Scales with Transformational Invariance.** To combine both locality at multiple spatial scales and transformational invariance, a kernel with locality at multiple spatial scales can be introduced as a base kernel into transformation-invariant kernels.

**Complexity Analysis and Memory Trade-off.** One may trade off between the memory requirement and computation time when it comes to the depth of the architecture. Supported by adequate memory space, one can store all kernel values from every layer, with both computation time and memory space increasing linearly with depth. In contrast, when limited by memory space, one can store only the kernel values from the final layer. In that case, although the memory requirement does not increase with depth, computation time grows exponentially with depth.

The time complexity of computing the polynomial kernel is between $O(n^2 m)$ and $O(n^3 m)$ based on LIBSVM [2], while space complexity is $O(n^2)$. With sufficient memory of order $O(n^2 m)$, the computations of kernel values can be parallelized so that the time complexity of the locality kernel is considerably reduced to between $O(n^2 kd)$ and $O(n^3 kd)$, where $k$ and $d$ are the locality parameter and the depth, respectively, with $kd \ll m$. Note that since our focus is on small sample sizes, the $O(n^2)$ complexity is acceptable.

Table 2: Original MNIST Dataset and EMNIST Letters Dataset (100, 200, 500 training samples): Test accuracy of newly proposed methods compared with the original SVM, the tangent distance (TD) nearest neighbors (two-sided), the RI-SVM based on Average Fit, and the best CNN. Based on the same training set, our fine-tuned ResNet achieves similar performance as in [10].

| Method | Original MNIST | | | EMNIST Letters | | |
| | 100 | 200 | 500 | 100 | 200 | 500 |
| | Acc/% | Acc/% | Acc/% | Acc/% | Acc/% | Acc/% |
|---|---|---|---|---|---|---|
| L-TI-RI-SVM | *81.55* | *89.23* | 92.58 | *44.56* | *55.18* | 66.42 |
| TI-RI-SVM | 75.10 | 86.47 | *93.11* | 43.16 | 52.40 | *67.42* |
| L-TI-SVM | 78.86 | 87.02 | 91.01 | 42.51 | 52.81 | 64.66 |
| L-RI-SVM | 77.96 | 83.96 | 89.65 | 38.39 | 47.29 | 59.76 |
| TI-SVM | 69.34 | 82.34 | 91.00 | 39.94 | 48.12 | 63.52 |
| RI-SVM | 73.82 | 83.60 | 90.19 | 38.03 | 45.02 | 59.04 |
| L-SVM | 75.27 | 82.11 | 88.21 | 37.01 | 45.08 | 58.05 |
| SVM | 68.16 | 78.67 | 87.14 | 36.65 | 42.74 | 56.38 |
| TD (two-sided) [30] | 73.04 | 81.68 | 88.15 | 37.99 | 45.31 | 57.63 |
| RI-SVM (Average-Fit) | 68.05 | 78.81 | 87.21 | 36.82 | 42.41 | 56.22 |
| Best CNN [10] / ResNet | 68.33 | 83.20 | 91.33 | 33.82 | 53.17 | 57.06 |

## 4 Experimental Evaluation

### 4.1 Datasets

We evaluate the performance of the methods developed on four datasets:

1. The Original MNIST Dataset [17]
2. The EMNIST Letters Dataset [4]
3. The Translated MNIST Dataset: Since most generally available datasets appear to have already been centered or otherwise preprocessed, we "uncenter" them to better verify the accuracy improvement of TI kernels. We place the objects in a larger (64*64*1) canvas, and then randomly translate them so that they are not necessarily centered but still maintain their integrity. In addition, we add a Gaussian noise ($\mu = 0, \sigma = 0.1$) to avoid being able to accurately center the image by calculating the center-of-mass. We call the resulting dataset the "Translated dataset". Figure 1(b) shows some samples from different classes of the Translated MNIST dataset.
4. The Rotated MNIST Dataset: We place the original image in the middle of a larger canvas and rotate it, with the blank area after rotation filled with Gaussian noise. RI kernels are not designed to and cannot distinguish between equivalent elements (e.g., 6 versus 9), and so we skip them. Figure 1(c) displays samples from different classes of the Rotated MNIST dataset.

### 4.2 Experimental Results and Observations

Table 2 and Figure 2 provide the test accuracy of all methods on the Original and Transformed MNIST Datasets, respectively, while Table 2 shows the test accuracy for the EMNIST Letters Dataset [4]. The letters L, TI, and RI represent Locality at multiple spatial scales, TI kernels, and RI kernels, respectively. A combination such as L-TI represents an SVM with both Locality and Translational Invariance. Note that the intended application of our proposed methods is to learn when data is scarce and there is no large database of similar data, which precludes the possibility of pre-training. We provide code at https://github.com/tliu1997/TI-SVM.

For the Original MNIST dataset with 100/200/500 training samples (Table 2), after introducing locality and transformational invariance, the classification accuracy is improved from 68.33%/83.20%/91.33% reported by the best CNNs optimized over architectures and dimensions [10] to 81.55%/89.23%/93.11% respectively. The improvements indicate that the original dataset does not center and deskew objects perfectly. Larger improvements can be observed on the EMNIST Letters dataset [4] compared with the original SVM, two-sided tangent distance (TD) nearest neighbors [30], RI kernel based on Average-Fit, and ResNet. (Since we find that the test accuracy of TD-based

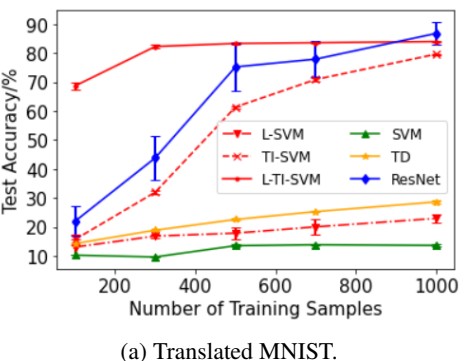
(a) Translated MNIST.

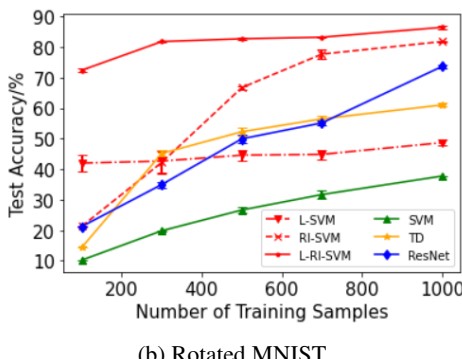
(b) Rotated MNIST.

Figure 2: Test accuracy vs. Number of training samples, for Transformed MNIST datasets. L-TI-SVM yields considerable improvement at small sizes in the case of translated samples, while, similarly, L-RI-SVM does so in the case of rotated samples.

nearest neighbors [30] is better than that of TD-based SVMs [13], we only report results of TD-based nearest neighbors as one of the baselines.) All results are multi-class classification accuracies.

In Figure 2, we present the obtained test accuracy as a function of the number of training samples, for two different transformed datasets. Experiments are performed 5 times with mean and standard deviation denoted by the length of the bars around the mean, for 100/300/500/700/1000 training samples respectively. (L-)TI-SVM and (L-)RI-SVM outperform ResNet in many cases when there is no data augmentation since they embed useful domain knowledge for classifiers, especially for the small size regime of training samples. However, with the increase in the number of training samples, the benefits brought by domain knowledge gradually decrease, as shown in Figure 2. Additionally, the test accuracy of the newly proposed methods has a smaller variance than ResNet's in general.

From the experimental results, we see that all SVMs with newly defined kernels improve upon the test accuracy of the original SVM method, whether they are original datasets or transformed datasets. They also greatly outperform the best CNNs in the small training sample regime of interest. For transformed datasets, improvements are more significant. Note that the performance improvement of proposed methods comes at the cost of longer computation time. When computation time is critical, we may simply use new single methods, i.e., L-SVM, TI-SVM, and RI-SVM, which enjoy a relatively good performance at a small cost of additional computation time.

### 4.3 Details of Experimental Evaluation

With consideration of computational speed and memory space, we utilize a two-layer structure (5) as well as a $\frac{k_1-1}{2}$-zero padding and a stride of 1 to implement locality. In order to compare the test accuracy of L-SVM, TI-SVM, RI-SVM and further combine them, we select a polynomial kernel with a fixed degree (8 in our experiments) to realize the proposed methods. Note that degree 8 is not necessarily the optimal degree; one can tune the specific degree for different datasets.

We compare our results with [10], which adopts a tree building technique to examine all the possible combinations of layers and corresponding dimensionalities to find the optimal CNN architecture. As for the DNN benchmark of the EMNIST Letters and the Transformed MNIST datasets, we select ResNet [14, 15], a classic CNN architecture, as a DNN benchmark. Plus, for fairness, we do not implement data augmentation for any models and train all from scratch.

Note that all experimental results are based on LIBSVM [2] and are carried out on an Intel Xeon E5-2697A V4 Linux server with a maximum clock rate of 2.6 GHz and a total memory of 512 GB.

## 5 Concluding Remarks

In this paper, we have developed transformation-invariant kernels that capture domain knowledge of the invariances in the domain. They can also additionally incorporate composition and locality at multiple spatial scales. The resulting kernels appear to provide significantly superior classification

performance in the small sample size regime that is of interest in this paper. Experiments demonstrate that for the same polynomial kernel, incorporating locality and transformational invariance improves accuracy, especially for situations where data is scarce.

## Acknowledgement

This material is based upon work by Texas A&M University that is partially supported by the US Army Contracting Command under W911NF-22-1-0151, US Office of Naval Research under N00014-21-1-2385, 4/21-22 DARES: Army Research Office W911NF-21-20064, US National Science Foundation under CMMI-2038625. The views expressed herein and conclusions contained in this document are those of the authors and should not be interpreted as representing the views or official policies, either expressed or implied, of the U.S. Army Contracting Command, ONR, ARO, NSF, or the United States Government. The U.S. Government is authorized to reproduce and distribute reprints for Government purposes notwithstanding any copyright notation herein.

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
