## A   Supporting Definitions and Lemmas

**Definition 1.** A random variable X with mean $\mu = \mathbb{E}[X]$ is *sub-exponential* if there are non-negative parameters $(\nu, b)$ such that [34]

$$\mathbb{E}[\exp(\lambda(X - \mu))] \leq \exp(\frac{\nu^2 \lambda^2}{2}), \ \ \forall |\lambda| < \frac{1}{b}.$$

We denote this by $X \sim SE(\nu, b)$.

**Lemma 4** (Sub-exponential tail bound [34]). *Suppose that $X \sim SE(\nu, b)$. Then,*

$$\mathbb{P}[|X - \mu| \geq t] \leq \begin{cases} 2\exp(-\frac{t^2}{2\nu^2}) & \text{if } 0 \leq t \leq \frac{\nu^2}{b}, \\ 2\exp(-\frac{t}{2b}) & \text{if } t > \frac{\nu^2}{b}. \end{cases}$$

Some properties related to sub-exponential random variables are listed below.

**Lemma 5** ([34]).

1. *For a standard normal random variable X, $X^2$ is $SE(2, 4)$;*

2. *For random variable $X \sim SE(\nu, b)$, $aX \sim SE(a\nu, ab)$;*

3. *Consider independent random variables $X_1, \ldots, X_n$, where $X_i \sim SE(\nu_i, b_i)$. Let $\nu = (\nu_1, \ldots, \nu_n)$ and $b = (b_1, \ldots, b_n)$. Then $\sum_{i=1}^n X_i \sim SE(\|\nu\|_2, \|b\|_\infty)$.*

## B   Proof of Theorem 1

**Theorem 3** (Restatement of Theorem 1). *Let*

$$K_{TI,best,linear}(X_i, X_j) := \sup_{T \in \mathcal{G}_{trans}} \frac{1}{m}(TX_i)^T X_j$$

*be the best-fit translation invariant kernel with the base kernel chosen as the normalized linear kernel. Under Assumption 1, if $n \leq \frac{\|\lambda\|_1}{2\|\lambda\|_2(\ln m)^{\frac{1+\epsilon}{2}}}$, then $K_{TI,best,linear}$ is a positive definite kernel with probability approaching one, as $m \to \infty$.*

*Proof of Theorem 1.* For brevity, we denote $K_{TI,best,linear}(X_i, X_j)$ and $K_{linear}(X_i, X_j)$ by $K_{TI,ij}$ and $K_{ij}$, respectively. From Gershgorin's circle theorem [33] every eigenvalue of $K_{TI,best,linear}$ lies within at least one of the Gershgorin discs $\mathcal{D}(K_{TI,ii}, r_i) := \{\lambda \in \mathbb{R} \mid |\lambda - K_{TI,ii}| \leq r_i\}$, where $r_i := \sum_{j \neq i} |K_{TI,ij}|$. Hence if $K_{TI,ii} > \sum_{j \neq i} |K_{TI,ij}|, \forall i$, then $K_{TI,best,linear}$ is a positive definite kernel.

Note that the dimension of the vector $X_i$ is $m = m_1 \times m_2$ in the case of a two-dimensional array. Under Assumption 1, $X_i \sim \mathcal{N}(0, \Sigma), \forall i \in [n]$. Since $\Sigma$ is symmetric and positive semi-definite, there exists an orthogonal matrix $O \in \mathbb{R}^{m \times m}$ with $\Sigma = O \cdot \text{diag}(\lambda^{(1)}, \ldots, \lambda^{(m)}) \cdot O^T$, where $\lambda^{(1)}, \ldots, \lambda^{(m)} \geq 0$ are the eigenvalues of $\Sigma$. Define $\Sigma^{\frac{1}{2}} := O \cdot \text{diag}(\sqrt{\lambda^{(1)}}, \ldots, \sqrt{\lambda^{(m)}}) \cdot O^T$, then $X_i = \Sigma^{\frac{1}{2}} Z_i$, where $Z_i \sim \mathcal{N}(0, I_m)$. Define $H_i := O^T Z_i \sim \mathcal{N}(0, I_m)$, then

$$\|X_i\|^2 = \left\| O \, \text{diag}(\sqrt{\lambda^{(1)}}, \ldots, \sqrt{\lambda^{(m)}}) O^T Z_i \right\|^2$$

$$= \left\| \text{diag}(\sqrt{\lambda^{(1)}}, \ldots, \sqrt{\lambda^{(m)}}) H_i \right\|^2 = \sum_{p=1}^m \lambda^{(p)}(H_i^{(p)})^2,$$

and $\mathbb{E}[\|X_i\|^2] = \sum_{p=1}^m \lambda^{(p)} \mathbb{E}[(H_i^{(p)})^2] = \|\lambda\|_1$. Let $\lambda := (\lambda^{(1)}, \ldots, \lambda^{(m)})$. Based on Lemma 5, we have $(H_i^{(p)})^2 \sim SE(2, 4)$, $\lambda^{(p)}(H_i^{(p)})^2 \sim SE(2\lambda^{(p)}, 4\lambda^{(p)})$, and $\|X_i\|^2 \sim SE(2\|\lambda\|_2, 4\|\lambda\|_\infty)$. According to Lemma 4,

$$\mathbb{P}\left[\frac{1}{m}\|X_i\|^2 \leq \frac{\|\lambda\|_1}{m} - \frac{t}{m}\right] \leq \begin{cases} \exp(-\frac{t^2}{8\|\lambda\|_2^2}) & \text{if } 0 \leq t \leq \frac{\|\lambda\|_2^2}{\|\lambda\|_\infty}, \\ \exp(-\frac{t}{8\|\lambda\|_\infty}) & \text{if } t > \frac{\|\lambda\|_2^2}{\|\lambda\|_\infty}. \end{cases}$$

Let $t = \|\lambda\|_1/2$, then we have

$$\mathbb{P}\left(K_{ii} \leq \frac{1}{2m}\|\lambda\|_1\right) \leq \max\left(\exp\left(-\frac{\|\lambda\|_1^2}{32\|\lambda\|_2^2}\right), \exp\left(-\frac{\|\lambda\|_1}{16\|\lambda\|_\infty}\right)\right)$$

$$\overset{(a)}{\leq} \exp\left(-\frac{\|\lambda\|_1}{32\|\lambda\|_\infty}\right),$$

where $(a)$ holds due to Holder's inequality. Noting that $K_{TI,ii} = \max_{T \in \mathcal{G}} K_{linear}(TX_i, X_i) \geq K_{ii}, \forall i$,

$$\mathbb{P}(K_{TI,ii} \leq \frac{1}{2m}\|\lambda\|_1) \leq \mathbb{P}(K_{ii} \leq \frac{1}{2m}\|\lambda\|_1) \leq \exp\left(-\frac{\|\lambda\|_1}{32\|\lambda\|_\infty}\right). \tag{7}$$

Now we turn to the off-diagonal terms $K_{TI,ij}$ for $i \neq j$. For $p \in [m]$, one can write

$$X_i^{(p)} X_j^{(p)} = (O\Lambda^{\frac{1}{2}}H_i)^T (O\Lambda^{\frac{1}{2}}H_j) = H_i^T \Lambda H_j = \sum_{p=1}^{m} \lambda^{(p)} H_i^{(p)} H_j^{(p)}.$$

Note that $H_i^{(p)} H_j^{(p)} = \frac{1}{2}(Y_{+,ij}^{(p)} - Y_{-,ij}^{(p)})(Y_{+,ij}^{(p)} + Y_{-,ij}^{(p)})$, where $Y_{+,ij}^{(p)} := \frac{1}{\sqrt{2}}(H_j^{(p)} + H_i^{(p)})$ and $Y_{-,ij}^{(p)} := \frac{1}{\sqrt{2}}(H_j^{(p)} - H_i^{(p)})$ are independent $N(0,1)$ random variables. Hence $(Y_{+,ij}^{(p)})^2$ and $(Y_{-,ij}^{(p)})^2$ are chi-squared random variables, and their moment generating functions are $\mathbb{E}[e^{r(Y_{+,ij}^{(p)})^2}] = \mathbb{E}[e^{r(Y_{-,ij}^{(p)})^2}] = \frac{1}{\sqrt{1-2r}}$ for $r < \frac{1}{2}$. Hence for any $|r| \leq 1/\|\lambda\|_\infty$, we know

$$\mathbb{E}[e^{rX_i^\top X_j}] = \mathbb{E}\left[\exp\left(r\sum_{p=1}^{m} \lambda^{(p)} H_i^{(p)} H_j^{(p)}\right)\right] = \mathbb{E}\left[\exp\left(\frac{r}{2}\sum_{p=1}^{m} \lambda^{(p)}\left((Y_{+,ij}^{(p)})^2 - (Y_{-,ij}^{(p)})^2\right)\right)\right]$$

$$\overset{(b)}{=} \prod_{p=1}^{m} \mathbb{E}\left[\exp(\frac{r\lambda^{(p)}}{2}(Y_{+,ij}^{(p)})^2)\right] \mathbb{E}\left[\exp(-\frac{r\lambda^{(p)}}{2}(Y_{-,ij}^{(p)})^2)\right]$$

$$= \prod_{p=1}^{m} \frac{1}{\sqrt{1 - (\lambda^{(p)})^2 r^2}},$$

where $(b)$ is true since the random variables $\{Y_{+,ij}^{(p)}, Y_{-,ij}^{(p)}\}_{p \in [m]}$ are mutually independent. It can be verified that $\frac{1}{\sqrt{1-x}} \leq e^x$ for $0 \leq x \leq 1/2$. We can then upper bound the moment generating function of $\mathbb{E}[e^{rX_i^\top X_j}]$, since for any $|r| \leq 1/\|\lambda\|_\infty$,

$$\mathbb{E}[e^{rX_i^\top X_j}] = \prod_{p=1}^{m} \frac{1}{\sqrt{1 - (\lambda^{(p)})^2 r^2}} \leq \prod_{p=1}^{m} \exp\left((\lambda^{(p)})^2 r^2\right) = \exp\left(\frac{2\|\lambda\|_2^2}{2} r^2\right).$$

This implies $X_i^\top X_j \sim SE(\sqrt{2}\|\lambda\|_2, \|\lambda\|_\infty)$. Since $\mathbb{E}[X_i^\top X_j] = 0$, by Lemma 4, we know

$$\mathbb{P}\left[X_i^\top X_j \geq t\right] \leq \begin{cases} \exp(-\frac{t^2}{4\|\lambda\|_2^2}) & \text{if } 0 \leq t \leq \frac{2\|\lambda\|_2^2}{\|\lambda\|_\infty}, \\ \exp(-\frac{t}{2\|\lambda\|_\infty}) & \text{if } t > \frac{2\|\lambda\|_2^2}{\|\lambda\|_\infty}. \end{cases}$$

Let $t = \|\lambda\|_2(\ln m)^{(1+\epsilon)/2}$. Under the assumption that $\frac{\|\lambda\|_2}{\|\lambda\|_\infty} \geq \frac{(\ln m)^{(1+\epsilon)/2}}{2}$, we have

$$\mathbb{P}\left[K_{ij} \geq \frac{\|\lambda\|_2(\ln m)^{\frac{1+\epsilon}{2}}}{m}\right] \leq \exp(-\frac{(\ln m)^{1+\epsilon}}{4}). \tag{8}$$

Note that $|K_{TI,ij}| = |\sup_T K_{linear}(TX_i, X_j)| \leq \sup_T |K_{linear}(TX_i, X_j)|$, then

$$\mathbb{P}(|K_{TI,ij}| \geq \frac{\|\lambda\|_2(\ln m)^{\frac{1+\epsilon}{2}}}{m}) \leq \mathbb{P}(\sup_T |K_{linear}(TX_i, X_j)| \geq \frac{\|\lambda\|_2(\ln m)^{\frac{1+\epsilon}{2}}}{m})$$

$$\overset{(c)}{\leq} m\mathbb{P}(|K_{ij}| \geq \frac{\|\lambda\|_2(\ln m)^{\frac{1+\epsilon}{2}}}{m}) \overset{(d)}{\leq} 2m\mathbb{P}(K_{ij} \geq \frac{\|\lambda\|_2(\ln m)^{\frac{1+\epsilon}{2}}}{m})$$

$$\leq \exp(-\frac{(\ln m)^{1+\epsilon}}{4} + \ln 2m), \tag{9}$$

where $(c)$ holds since the distribution is translation invariant, and $(d)$ holds since $K_{ij}$ is a symmetric random variable, i.e., $p_{K_{ij}}(y) = p_{K_{ij}}(-y)$. Combining (7) and (9), we have

$$\mathbb{P}(K_{TI,ii} > \sum_{j \neq i} |K_{TI,ij}|, \forall i) \geq \mathbb{P}(\min_i K_{TI,ii} > \max_i \sum_{j \neq i} |K_{TI,ij}|)$$

$$\geq \mathbb{P}(\min_i K_{TI,ii} > \frac{\|\lambda\|_1}{2m} \text{ and } \max_i \sum_{j \neq i} |K_{TI,ij}| < \frac{\|\lambda\|_2 (\ln m)^{\frac{1+\epsilon}{2}} n}{m})$$

$$= 1 - \mathbb{P}(\min_i K_{TI,ii} \leq \frac{\|\lambda\|_1}{2m} \text{ or } \max_i \sum_{j \neq i} |K_{TI,ij}| \geq \frac{\|\lambda\|_2 (\ln m)^{\frac{1+\epsilon}{2}} n}{m})$$

$$\geq 1 - \mathbb{P}(\min_i K_{TI,ii} \leq \frac{\|\lambda\|_1}{2m}) - \mathbb{P}(\max_i \sum_{j \neq i} |K_{Ti,ij}| \geq \frac{\|\lambda\|_2 (\ln m)^{\frac{1+\epsilon}{2}} n}{m})$$

$$= 1 - \mathbb{P}(K_{TI,ii} \leq \frac{\|\lambda\|_1}{2m}, \exists i) - \mathbb{P}(\sum_{j \neq i} |K_{Ti,ij}| \geq \frac{\|\lambda\|_2 (\ln m)^{\frac{1+\epsilon}{2}} n}{m}), \exists i)$$

$$\overset{(e)}{\geq} 1 - n\mathbb{P}(K_{TI,ii} \leq \frac{\|\lambda\|_1}{2m}) - n\mathbb{P}(|K_{Ti,ij}| \geq \frac{\|\lambda\|_2 (\ln m)^{\frac{1+\epsilon}{2}}}{m}, \exists j \neq i)$$

$$\geq 1 - \exp(-\frac{\|\lambda\|_1}{32\|\lambda\|_\infty} + \ln n) - \exp(-\frac{1}{4}(\ln m)^{1+\epsilon} + 2\ln n + \ln 2m).$$

Above, $(e)$ holds since the probability distributions of $|K_{TI,ij}|$ are identical for all $j \neq i$. Since $\|\lambda\|_1 \geq \|\lambda\|_2$, the assumption $\frac{\|\lambda\|_2}{\|\lambda\|_\infty} \geq (\ln m)^{(1+\epsilon)/2}$ implies $\frac{\|\lambda\|_1}{\|\lambda\|_\infty} \geq (\ln m)^{(1+\epsilon)/2}$. Note that since $n = \tilde{O}(\sqrt{m})$,

$$\lim_{m \to \infty} \mathbb{P}(K_{TI,ii} > \sum_{j \neq i} |K_{TI,ij}|, \forall i) \geq \lim_{m \to \infty} 1 - \frac{1}{poly(m)} = 1.$$

Therefore, $K_{TI,best,linear}$ is a positive definite kernel with probability approaching one as $m \to \infty$. $\qquad\square$

## C  Proof of Theorem 2

**Theorem 4** (Restatement of Theorem 2). *For any $\gamma \in \mathbb{R}_+$ and $d \in \mathbb{N}$, the translation-invariant kernels,*

$$K_{TI,best,poly}(X_i, X_j) := \sup_{T \in \mathcal{G}_{trans}} (1 + \frac{\gamma}{m}(TX_i)^T X_j)^d$$

*are positive definite with probability approaching 1 as $m \to +\infty$, under Assumption 1, $\|\lambda\|_\infty \leq 1$, and $n \leq \frac{\|\lambda\|_1}{2\|\lambda\|_2 (\ln m)^{\frac{1+\epsilon}{2}}}$.*

*Proof.* Define event $A_{\gamma,m,n} := \{\frac{\gamma}{m}(TX_i)^\top X_j \geq -1, \forall T \in \mathcal{G}, \forall i \neq j\}$, and event $B_{m,n} := \{K_{TI,best,linear} \text{ is } pd\}$. Denote $K(\cdot, \cdot) = (1 + \gamma K_{TI,best,linear}(\cdot, \cdot))^d$. Conditioned on event $A_{\gamma,m,n}$, $K_{TI,best,poly}(X_i, X_j) = K(X_i, X_j)$ for off-diagonal entries, and $K_{TI,best,poly}(X_i, X_i) \geq K(X_i, X_i)$. This implies $K_{TI,best,poly} \succeq K$. Conditioned on event $B_{m,n}$, the three properties in the proof of Lemma 3 indicate $K$ is pd. Thus $K_{TI,best,poly}$ is pd conditioned on $A_{\gamma,m,n} \cap B_{m,n}$.

Now $B_{m,n}$ holds w.h.p. by Theorem 1. By the symmetric distribution of $K_{ij}$ and (8), we have

$$\mathbb{P}(\frac{\gamma}{m}(X_i)^T X_j \leq -\frac{\gamma \|\lambda\|_2 (\ln m)^{\frac{1+\epsilon}{2}}}{m}) \leq \exp(-\frac{(\ln m)^{1+\epsilon}}{4}).$$

Due to $\|\lambda\|_2 \leq \sqrt{m}$ and union bounds, $\lim_{m \to \infty} \mathbb{P}(A_{\gamma,m,n}) \geq \lim_{m \to \infty} 1 - \frac{1}{poly(m)} = 1.$ $\qquad\square$

# D Extensions to other kernels

The property of transformational invariance can be extended to other kernels with the guarantee of positive definiteness, e.g., radial basis function (RBF) kernels with norm-preserving transformations (i.e., $\|TX_i\| = \|X_i\|, \forall i \in [n]$). Define the normalized base kernel as

$$K_{RBF}(X_i, X_j) := \exp(-\frac{\|X_i - X_j\|^2}{2m\sigma^2}) = \exp(-\frac{\|X_i\|^2 + \|X_j\|^2}{2m\sigma^2}) \exp(\frac{X_i^T X_j}{m\sigma^2}).$$

If transformations are norm-preserving, then

$$K_{TI,best,RBF}(X_i, X_j) := \exp(-\frac{\|X_i\|^2 + \|X_j\|^2}{2m\sigma^2}) \sup_{T \in \mathcal{G}_{trans}} \exp(\frac{(TX_i)^T X_j}{m\sigma^2})$$

$$= \exp(-\frac{\|X_i\|^2 + \|X_j\|^2}{2m\sigma^2}) \exp(\sup_{T \in \mathcal{G}_{trans}} \frac{(TX_i)^T X_j}{m\sigma^2}).$$

By Theorem 1 and three properties in the proof of Lemma 3, $K_{TI,best,rbf}$ is still positive definite since $e^x = \sum_{i=0}^{\infty} \frac{x^i}{i!}$. However, the design of locality at multiple scales doesn't apply to RBF kernels since the kernel tricks cannot be utilized anymore. Since we merge two designs in the experimental part, we choose polynomial kernels to illustrate the designs in the paper.