# OpenReview forum: "Learning from Few Samples: Transformation-Invariant SVMs with Composition and Locality at Multiple Scales"
_NeurIPS.cc/2022/Conference — NeurIPS 2022 Accept_

### Official Review · Reviewer_R7kt · 2022-07-02

**Rating:** 7
**Confidence:** 2
**Soundness:** 3 good
**Presentation:** 3 good
**Contribution:** 3 good

**Summary:**

** I thank the reviewers for their answers **

The paper analysis theoretically the best fit kernel under transformational invariance. Such kernel is not necessarily definite positive. The authors prove some bounds for which it is with hight probability as long as the sample size is small enough.

The authors also incorporate into the kernel the knowledge of locality.

This strategy is inspired from the success of deep convolutional neural networks.

To my understanding, the paper focuses on small samples mainly as a limitation rather than as a specific objective (this is not a critic).

The experiments confirme the benefit of the method on small samples.

**Questions:**

I understand that in the experiments L is locality, TI is translational invariance and RI rotational invariance but I think that it would improve readability to introduces the notations.

Could you provide an empirical curve of probability to be positive definite given the sample size, maybe on a syntetic dataset ?

Could you provide std on your experiments. Small samples might be linked to large variance in the experiments.

**Limitations:**

Some limitations are given in the core of the paper but not specifically addressed.

**Strengths And Weaknesses:**

- Strengths
The paper provides some theoretical and practical elements about the use of such kernels for small samples. They show that even though such kernels are not always positive definite, they can be as such on small samples.

- Weaknesses
The paper is a succession of many elements that I found some time difficult to ingest.
Experiments do show std which might be large on small samples.

---

> ### Author Response · Authors · 2022-08-02
> **Response to Reviewer R7kt (1/2)**
>
> We thank the reviewer for the positive review and constructive feedback.
>
> 1.  "To my understanding, the paper focuses on small samples mainly as a limitation rather than as a specific objective (this is not a critic)."
>
> *Response:* We respectfully disagree with this. As shown in the title and abstract, the specific objective of the paper is to learn from small samples. With this specific objective in mind, we incorporate prior knowledge about images into SVMs since prior knowledge is especially useful in small samples. This is motivated by how CNNs have incorporated similar prior knowledge for better recognition. We analyze the critical property of positive definiteness for the resulting kernels.
>
> 2.  *"The paper is a succession of many elements that I found some time difficult to ingest."*
>
> *Response:* To enhance as of understanding digestion, we will explain the intuitions behind key results as follows. When designing a new transformation-invariant kernel, we first need to verify that it is indeed symmetric and transformation-invariant (i.e., Lemma 1). We then examine when the new kernel is positive-definite since only then are there provable guarantees of performance. The major theoretical results concerning positive definiteness are in Theorems 1 and 2 and Lemma 2 under Assumption 1. To further take advantage of prior knowledge, we show how one may incorporate locality at multiple spatial scales resulting in new kernels, and show that the property of positive definiteness continues to hold (i.e., Lemma 3). Then we utilize four different datasets to empirically verify that the proposed methods do provide superior accuracy in comparison to other baselines for small sample sizes.
>
> 3.  *"...improve readability to introduce the notations."*
>
> *Response:* We had explained the notations in Lines 289-292. To further improve readability, we will explain the notations again in the titles of related tables and figures.
>
> 4.  *"...provide an empirical curve of probability to be positive definite given the sample size, maybe on a synthetic dataset?"*
>
> *Response:* We use $K_{TI, best, linear}$ as an example and present the probability of positive definiteness for a given sample size in the following table. For the synthetic dataset, we generate i.i.d. samples with i.i.d. features from the standard normal distribution, where the number of features is 784. To estimate the probability of positive definite, we run experiments 10 times (by randomly generating 10 different datasets with the same distribution) to check if all eigenvalues of the transformation-invariant kernel matrix are greater than zero. We particularly focus on values of $n$ where there appears to be phase transition. This further points to the conservativeness of Gershgorin's method.
>
>       Method                     750   775   800   810   825   850
>       ------------------------ ----- ----- ----- ----- ----- -----
>       $K_{TI, best, linear}$     1.0   1.0   0.8   0.5   0.1   0.0
>
> 5.  *" Some limitations are given in the core of the paper but not specifically addressed."*
>
> *Response:* As the reviewer observes, we had mentioned the limitation of this paper in Lines 168-171 concerning the conservativeness of the bound on the number of samples for positive definiteness. This limitation comes from the conservative bound of Gershgorin's circle theorem, which is a mathematical challenge and an open problem for future work. Another limitation concerns the computation time of SVMs in general as the number of samples grows.

---

> > ### Author Response · Authors · 2022-08-02
> > **Response to Reviewer R7kt (2/2)**
> >
> > 6.  *" provide std on the experiments."*
> >
> > *Response:* We had shown error bars (standard deviation) in Figure 2, but now provide the specific numerical values in parentheses of the following tables. The tables below demonstrate that SVM-based methods enjoy a smaller standard deviation compared with DL methods (ResNet). (The letters L, TI, RI, and TD represent locality at multiple spatial scales, transformation-invariant kernels, rotation-invariant kernels, and tangent distance kernels, respectively.)
> >
> >       Method              100          300          500          700         1000
> >       ---------- ------------ ------------ ------------ ------------ ------------
> >       L-TI-SVM     68.7 (1.4)   82.3 (0.7)   83.3 (0.4)   83.6 (0.2)   85.0 (0.2)
> >       TI-SVM       15.6 (1.9)   32.0 (0.5)   61.4 (0.4)   70.9 (0.5)   79.7 (0.2)
> >       L-SVM        13.0 (2.7)   16.8 (0.7)   17.8 (2.1)   20.0 (2.5)   22.9 (1.3)
> >       SVM          10.2 (0.0)    9.6 (0.0)   13.5 (0.3)   13.8 (0.3)   13.6 (0.4)
> >       TD           14.2 (0.2)   18.8 (0.3)   22.5 (0.4)   25.3 (0.3)   28.6 (0.7)
> >       ResNet       17.7 (4.0)   63.7 (5.7)   69.2 (5.7)   76.4 (5.8)   83.7 (4.5)
> >
> >       Method              100          300          500          700         1000
> >       ---------- ------------ ------------ ------------ ------------ ------------
> >       L-RI-SVM     72.4 (0.7)   81.8 (0.1)   82.7 (0.5)   83.2 (0.3)   86.5 (0.5)
> >       RI-SVM       21.3 (0.1)   42.2 (3.6)   66.7 (0.6)   77.7 (1.5)   81.8 (0.1)
> >       L-SVM        42.0 (2.7)   42.7 (4.6)   44.6 (1.8)   44.9 (1.5)   48.7 (1.0)
> >       SVM          10.2 (0.0)   19.8 (0.2)   26.6 (1.1)   31.7 (1.3)   37.7 (0.1)
> >       TD           14.3 (0.1)   45.2 (1.0)   52.3 (1.5)   56.5 (0.9)   61.1 (0.6)
> >       ResNet       11.7 (1.2)   17.7 (4.5)   51.7 (5.8)   54.0 (5.1)   63.5 (2.5)

---

### Official Review · Reviewer_wPUn · 2022-07-15

**Rating:** 7
**Confidence:** 3
**Soundness:** 4 excellent
**Presentation:** 4 excellent
**Contribution:** 3 good

**Summary:**

The paper proposes a novel formulation of SVM to tackle the problem of learning in scarce data regime. it is inspired by the properties of CNNs, namely translation invariance and locality at different scales.

**Questions:**

1/ What is the training and inference time of the proposed method? it could be interesting to add a column mentioning the training time & inference time and the number of parameters when comparing with related works and DL methods.

2/ For the sake of comparison and a better understanding of the formulation of the proposed method, could you make some visualizations of the learned kernel as in CNNs to show the learned filters? (figure 3 in ImageNet Classification with Deep Convolutional
Neural Networks https://proceedings.neurips.cc/paper/2012/file/c399862d3b9d6b76c8436e924a68c45b-Paper.pdf)

**Limitations:**

This work doesn't have any negative societal impact.
The authors didn't mention the limitations of their works.

**Strengths And Weaknesses:**

The success of CNNs is related to the availability of large-scale data and the development of hardware. Moreover, the properties of CNNs such as compositionality, translation & rotation invariance,  locality, and multi-scale representations have made CNNs successful in a large scope of applications. However, CNN's are data hungrier. In this paper, the authors tackle the problem of learning in a small data regime by reformulating SVMs to take into consideration the translation invariance, locality, and compositional structure properties of CNNs.
One of the strengths of the paper is in the modeling of the proposed SVM and the mathematical validations,  including the verification of the satisfactions of the positive-semi definiteness of the proposed kernel.  The claims are well supported and the mathematical variables are well defined which makes the paper easy to follow.

The methodological part includes two theorems and three lemmas that are validated empirically considering different experimental protocols. However, the experimental part could have included other applications (video, speech, physical data) where the data are rare and tasks (regression, etc) to show the versatility of the methods.

---

> ### Author Response · Authors · 2022-08-02
> **Response to Reviewer wPUn (1/2)**
>
> We thank the reviewer for the positive review and constructive feedback.
>
> 1.  *"Experimental part could have included other applications (video, speech, physical data)."*
>
> *Response:* Previously, we worked with collaborators from the department of physics to adopt the proposed methods for jet image classification of light quarks and gluons from simulated particle collisions. The jet images simulate the angular distribution of the energy carried by the spray of an initial light quark or gluon. We trained all methods based on the same 100 training samples and report below the test accuracy on 10,000 hold-out test samples. Due to the toroidal imaging surface, the images have both translational and rotational invariance. Since pixel values are sparse in jet images and adjacent pixels don't have strong correlations as traditional images, we only report results about transformation-invariant kernels and several baselines in the following table. The proposed kernels enjoy higher test accuracy compared with other methods. (The letters TI, RI, and TD represent transformation-invariant kernels, rotation-invariant kernels, and tangent distance kernels, respectively.) We will try to seek permission from physics collaborators to make the dataset public in our final version. We believe that our proposed methods can also work for other applications as long as the prior knowledge (transformational invariance and locality) is satisfied.
>
>       Method         TI-RI-SVM   TI-SVM   RI-SVM     SVM      TD   ResNet
>       ------------ ----------- -------- -------- ------- ------- --------
>       Test-acc/%         93.64    92.92    91.25   79.64   81.97    85.87
>
> 2.  *"make some visualizations of the learned kernel as in CNNs to show the learned filters?"*
>
> *Response:* We will include it in our final version.
>
> 3.  *"The authors didn't mention the limitations of their works."*
>
> *Response:* As pointed out in the checklist, we listed the limitations of our work in Lines 168-171 about the conservativeness of the bound given by Gershgorin's circle theorem. Additionally, we will mention the comparison of computation time clearly in our final version.

---

> > ### Author Response · Authors · 2022-08-02
> > **Response to Reviewer wPUn (2/2)**
> >
> > 4.  *"...add a column mentioning training time & inference time and the number of parameters when comparing with related works and DL methods."*
> >
> > *Response:*    Since all SVM-based methods can take advantage of kernel tricks and solve the dual problem, the number of parameters of these methods is equal to the number of samples, i.e., 100-1000 in this paper. Additionally, given that the backbone of DL methods is ResNet-18, the number of parameters is about 11 million.
> >
> > Since listing all additional information for all methods and datasets is space-consuming, we present all information for all methods in the original MNIST dataset with 100 training samples. Note that we report the training time as the time required to train 100 training samples, while the inference time refers to the time required to infer an unseen sample. (Note that the inference time of a sample includes the call time of the trained model, so the total inference time of 10,000 samples is much less than 10,000 times the reported time. For example, the total inference time of TI-RI-SVM and ResNet for 10,000 test samples is 237.7 and 224.3 seconds respectively.) As shown in the following table, compared with the original SVM, the new SVM-based methods enjoy a much better test accuracy performance at the cost of longer training and inference time. The additional computation cost is caused by sequentially handling different transformations with the locality structure, which could be alleviated by techniques of parallel computing and GPU acceleration. However, the goal of this paper is to theoretically and empirically illustrate that incorporating into SVMs those properties that have made CNNs successful can enjoy better test accuracy for the problem of learning with small sample sizes. The acceleration of the proposed new methods is an engineering problem that we plan to solve in the future and we will mention it clearly in our final version. Thank you for pointing it out. Additionally, when computation time is critical, we may simply use new single methods, i.e., L-SVM, TI-SVM, and RI-SVM, which enjoy a relatively good performance at a small cost of additional computation time.
> >
> >       Method                   train-time/s   infer-time/s   \# of params   test-acc/%
> >       ---------------------- -------------- -------------- -------------- ------------
> >       ResNet                          728.2           38.4     11 million        68.33
> >       L-TI-RI-SVM                     699.8          199.3            100        81.55
> >       TI-RI-SVM                       123.9           57.0            100        75.10
> >       L-TI-SVM                         56.7           14.5            100        78.86
> >       L-RI-SVM                          2.4            0.7            100        77.96
> >       TI-SVM                            0.2            0.1            100        69.34
> >       RI-SVM                            0.6            0.3            100        73.82
> >       L-SVM                             0.3            0.1            100        75.27
> >       SVM                               0.1           0.01            100        68.16
> >       TD (two-sided)                    4.0            0.1            100        73.04
> >       RI-SVM (Average-Fit)              0.6            0.3            100        68.05

---

### Official Review · Reviewer_gSiY · 2022-07-20

**Rating:** 6
**Confidence:** 4
**Soundness:** 3 good
**Presentation:** 3 good
**Contribution:** 2 fair

**Summary:**

The authors propose a novel approach to incorporate into SVMs several properties seen in CNNs. In a specific way, authors introduce domain knowledge of invariances (i.e. image recognition), maximum similarity on a group of transformation (kernel based), incorporating local features at multiple spatial scales, as well as composition through the architecture of multiple layers. This characteristics are shown in a set of experiments wit the polynomial kernel and the L-TI-RI-SVM (and variants), CNN and ResNet applied into MNIST, EMNIST and translated MNIST.

**Questions:**

•	The manuscript section organization is missing from the introduction, it should be included.
•	Is it possible to consider to show results in table 2 in a visual manneer (i.e. figure 2)?
•	Is it possible to extend the work on future directions as, per example, other kernels?


**Limitations:**

The authors address the problem and found that polynomial kernels, under certain conditions can address the problem in image recognition. It will be nice to have the code in order to reproduce results and the corresponding architectures.

**Strengths And Weaknesses:**

Strengths: The core idea of the work is original in the sense of incorporating main characteristics involved in CNNs, into the construction of SVMs. Taking advantage of kernels, authors established the conditions for a symmetric kernel that is positive-definite transformation-invariant (polynomial kernel). The locality at multiple special spaces is introduced as well in the kernel with these characteristics. Figure  2 explains the advantages of the L-TI-SVM and L-RI-SVM in the T-MNIST and R-MINST

Weaknesses: the contribution on the is, at this stage, involved in problems related to image recognition, involved with the structure of pixels (and all the inner related characteristics). It is important to remark that authors use a very well-known dataset; MNIST and some variants. It is also important to consider that other datasets are also used as benchmarks, tackling few shot learning in images. It should be nice to offer a github link in order to reproduce the results.

---

> ### Author Response · Authors · 2022-08-02
> **Response to Reviewer gSiY**
>
> We thank the reviewer for the positive review and constructive feedback.
>
> 1. *``The manuscript section organization is missing from the introduction.''*
>
> *Response:* We will add the following paragraph in the Introduction. ``The remainder of this paper is organized as follows: kernels with transformational invariance (including definition, analysis of positive definiteness, comparison with other methods) are introduced in Section 2; kernels with locality at multiple scales (including definition and corresponding analysis) are provided in Section 3; experimental results are presented in Section 4; concluding in Section 5.''
>
> 2. *``... consider to show results in Table 2 in a visual manner.''*
>
> *Response:* We will visualize it in our final version.
>
> 3. *... extend the work on future direction, e.g., other kernels?*
>
> *Response:* The property of transformational invariance can be extended to other kernels with the guarantee of positive definiteness, e.g., radial basis function (RBF) kernels with norm-preserving transformations (i.e., $\\|T X_i\\| = \\|X_i\\|, \forall i \in [n]$). Define the normalized base kernel as
> \begin{align*}
> K_{RBF}(X_i, X_j) := \exp(-\frac{\\|X_i - X_j\\|^2}{2 m \sigma^2}) = \exp(-\frac{\\|X_i\\|^2 + \\|X_j\\|^2}{2 m \sigma^2}) \exp(\frac{X_i^T X_j}{ m \sigma^2}).
> \end{align*}
> If transformations are norm-preserving, then
> \begin{align*}
> K_{TI, best, RBF}(X_i, X_j) = \exp(-\frac{\\|X_i\\|^2 + \\|X_j\\|^2}{2 m \sigma^2}) \exp(\sup_{T \in \mathcal{G}_{trans}} \frac{(T X_i)^T X_j}{ m \sigma^2}).
> \end{align*}
>
> By Theorem 1 and three properties in the proof of Lemma 3, $K_{TI, best, rbf}$ is still positive definite since $e^x = \sum_{i=0}^{\infty} \frac{x^i}{i!}$. However, the design of locality at multiple scales doesn't apply to RBF kernels since the kernel tricks cannot be utilized anymore. Since we merge two designs in the experimental part, we choose polynomial kernels to illustrate the designs in the paper.
>
> 4.  *"\... consider other datasets are also used as benchmarks, tackling few-shot learning in images."*
>
> *Response:* Our task is different from few-shot learning. Few-shot learning consists of pre-training several related tasks during the meta-training phase, so that it can generalize well to unseen tasks with just a few examples. However, the intended application of our proposed method is to learn when data is scarce and there is no large database of similar data, which precludes the possibility of pre-training.
>
> As suggested by reviewer wPUn, we include below additional results about additional data, which classifies jet images of light quarks and gluons from simulated collisions. The jet images ($30 \times 30$ pixel) simulate the angular distribution of the energy carried by the spray of an initial light quark or gluon. We trained all methods based on the same 100 training samples and report below the test accuracy on 10,000 hold-out test samples. Due to the toroidal imaging surface, the images have both translational and rotational invariance. Since pixel values are sparse in jet images and adjacent pixels don't have strong correlations as traditional images, we only report results about transformation-invariant kernels and several baselines in the following table. The proposed kernels enjoy higher test accuracy compared with other methods. (The letters TI, RI, and TD represent transformation-invariant kernels, rotation-invariant kernels, and tangent distance kernels, respectively.) We will try to seek permission from physics collaborators to make the dataset public in our final version. We believe that our proposed methods can also work for other applications as long as the prior knowledge (transformational invariance and locality) is satisfied.
>
>       Method         TI-RI-SVM   TI-SVM   RI-SVM     SVM      TD   ResNet
>       ------------ ----------- -------- -------- ------- ------- --------
>       Test-acc/%         93.64    92.92    91.25   79.64   81.97    85.87
>
> 5.  *"...offer a GitHub link in order to reproduce the results."*
>
> *Response:* We had already included all codes to reproduce the results in the supplementary material. We will provide a public GitHub link in the final version.

---

### Meta-Review · Area_Chair_XmiY · 2022-08-31

**Recommendation:** Accept
**Confidence:** Certain

**Metareview:**

All reviews are quite positive.

Accept.

**Award:**

No

---

### Decision · Program_Chairs · 2022-09-14

Accept